# Phylogenomic Analysis of Two Species of *Parasenecio* and Comparative Analysis within Tribe Senecioneae (Asteraceae)

**Xiaofeng Liu [1,2], Junjia Luo [2], Mingke Zhang [2], Qiang Wang [3], Jie Liu [3], Die Wu [3,\*] and Zhixi Fu [1,2,4,\*]**

1   Key Laboratory of Land Resources Evaluation and Monitoring in Southwest, Sichuan Normal University, Ministry of Education, Chengdu 610066, China
2   College of Life Sciences, Sichuan Normal University, Chengdu 610101, China
3   School of Computer Science, Sichuan Normal University, Chengdu 610101, China
4   Sustainable Development Research Center of Resources and Environment of Western Sichuan, Sichuan Normal University, Chengdu 610101, China
\*   Correspondence: wd@sicnu.edu.cn (D.W.); fuzx2017@sicnu.edu.cn (Z.F.); Tel.: +86-18188309455 (D.W.); +86-13219011396 (Z.F.)

**Abstract:** The genus of *Parasenecio* (Senecioneae) comprises about 70 species of high medicinal value. In this study, the plastomes of *Parasenecio palmatisectus* and *P. latipes* were newly sequenced using high-throughput sequencing technology and compared with those of eight other species in Senecioneae. The complete chloroplast (cp) genomes were 151,185 bp in *P. latipes* with 37.5% GC and 151,263 bp in *P. palmatisectus* with 37.5% GC. We predicted 133 genes, including 37 tRNA genes, 86 protein-coding genes, 8 rRNA genes, and 2 pseudogenes (*ycf1* and *rps19*). A comparative genomic analysis showed that the complete cp genome sequences of *Parasenecio* species and their related species were relatively conserved. A total of 49 to 61 simple sequence repeats (SSRs) and 34 to 46 interspersed repeat sequences were identified in the 10 Senecioneae species of plastomes. Within the tribe Senecioneae, single-copy regions were more variable than inverted repeats regions, and the intergenic regions were more variable than the coding regions. Two genic regions (*ycf1* and *ccsA*) and four intergenic regions (*trnC-GCA-petN*, *ycf1-trnN-GUU*, *psaI-ycf4*, and *rpl32-trnL-UAG*) were identified as highly valuable plastid markers. A phylogenetic analysis under maximum likelihood revealed that the two *Parasenecio* species are sister to the genera of *Ligularia* and *Sinosenecio* in the tribe Senecioneae. This study also contributes to the super-barcode, phylogenetic, and evolutionary studies of *Parasenecio* plants.

**Keywords:** *Parasenecio*; chloroplast genome; comparative genomics; plastome phylogeny

## 1. Introduction

The genus *Parasenecio* Smith & Small belongs to the tribe Senecioneae of the family Asteraceae. There are about 70 species [1–6] that mainly occur in East Asia and the Sino-Himalayan region [2], and about 60 species in China [2]. *Parasenecio* is divided into six sections according to the leaf morphology and chromosome numbers [2]. Previous molecular phylogenetic studies showed that it is polyphyletic and forms a complex with other Eastern Asian genera of Tussilagininae, such as the *Ligularia–Cremanthodium–Parasenecio* complex (LCP) [7–12].

*Parasenecio* species have long been used as traditional Chinese medicines due to their significant bioactivities, such as their anti-inflammatory, antitussive, and antimicrobial properties [13,14]. Former studies of *Parasenecio* species mainly comprised medicinal and pharmacological studies [13,14]. However, few studies have reported the plastid genomes and phylogenetic analysis of species of *Parasenecio*. Genomic resources need to be developed to better utilize these medicinal plants [15–17]. Furthermore, a good knowledge of the genomic information of these medicinal species could inform biodiversity conservation efforts [18]. The species of *P. palmatisectus* is distributed in Sichuan, Xizang,

and Yunnan provinces in China and in Bhutan. The species of *P. latipes* is endemic to China and is only distributed in Yunnan and Sichuan provinces. These species are used for the treatment of inflammation and infectious diseases in Chinese folk medicine [2]. The medicinal importance of these two *Parasenecio* species makes their genetic and phylogenetic characterization more important.

The cp genome is a photosynthetic organelle in plants [19] that has an independent genome. The genome is typically 120 to 160 kb in size, containing 120 to 140 genes [20]. In general, the cp genome is a quadripartite DNA molecule. It includes a short single-copy (SSC) region, a large single-copy (LSC) region, and two inverted repeats (IRa and IRb) [21]. Compared with nuclear genomes, cp genomes are more conservative in gene order, content, and structure [21–24]. Therefore, they can be used as an effective tool for revealing phylogenetic relationships [22,25,26] and a super-barcode for identifying closely related species [22,27,28]. With the rapid development of sequencing technologies, the cp genome has been successfully used to infer phylogenetic relationships of Asteraceae, and it also provided a good opportunity to study the structural features, variation, and evolution of Asteraceae [20,22–27,29–35]. However, the complete cp genomes of *Parasenecio* have not been sequenced, which hampers the study of the evolution and phylogenetics of *Parasenecio*.

In this study, we assembled the complete cp genomes of *P. palmatisectus* and *P. latipes* and carried out a comparative plastome analysis with eight other published cp genomes of closely related species to detect the differences in the cp genome. Our research purposes were to (1) study the cp genomic features of *Parasenecio* species, (2) determine the structural variation of *Parasenecio* species by comparing the cp genomes with eight other species of the tribe Senecioneae, and (3) reconstruct the phylogenetic relationship of *Parasenecio* species and its related genera. Our results provide valuable genomic information for further studies on the phylogenetic relationships and sustainable utilization of *Parasenecio* species.

## 2. Materials and Methods

### 2.1. Plant Material, DNA Extraction, and Sequencing

Fresh leaves of *P. latipes* and *P. palmatisectus* were collected from Muli county and Bazhong city in Sichuan province, China, respectively (Figure 1). The fresh leaves were cleaned using 75% alcohol and ddH$_2$O, rapidly placed in liquid nitrogen, and then transferred to −80 °C for storage after returning to the laboratory. The voucher specimen of *P. latipes* (no. FZX5888) and the voucher specimen of *P. palmatisectus* (no. DY122) were deposited under the herbarium of Sichuan Normal University (SCNU), Chengdu city, Sichuan Province, China (contact: Dr. Prof. Zhixi Fu, fuzx2017@sicnu.edu.cn). The total genomic DNA was isolated from silica-dried leaves using a modified cetyltrimethylammonium bromide (CTAB) method [36]. The library construction and whole-genome sequencing were performed using high-throughput Illumina sequencing technology at Beijing Genomics Institute (Shenzhen, Guangdong, China, BGI). The qualified library was sequenced with the BGI NovaSeq 6000 platform, with a sequencing read length of 150 bp.

### 2.2. CP Genome Assembly and Annotation

The cp genomes were assembled using the SPAdes software (v3.10.1) with default settings [37]. The circular maps were identified using Bandage so as to assess the quality of the assembly [38]. The plastomes were annotated using Plastid Genome Annotator (PGA) with the cp genome sequence of *Senecio vulgaris* L. (NC_046693) [39] as the reference. The annotation results were inspected using Geneious R11 [40] and adjusted manually as needed. Finally, Organellar Genome Draw (OGDraw) (https://chlorobox.mpimp-golm.mpg.de/OGDraw.html (accessed on 6 June 2022)) was used to draw circular maps of the plastome [41]. The assembled complete cp genome sequences were submitted to NCBI GenBank (https://www.ncbi.nlm.nih.gov (accessed on 12 June 2022)) with the accession numbers ON749759 (*P. latipes*) and ON749760 (*P. palmatisectus*).

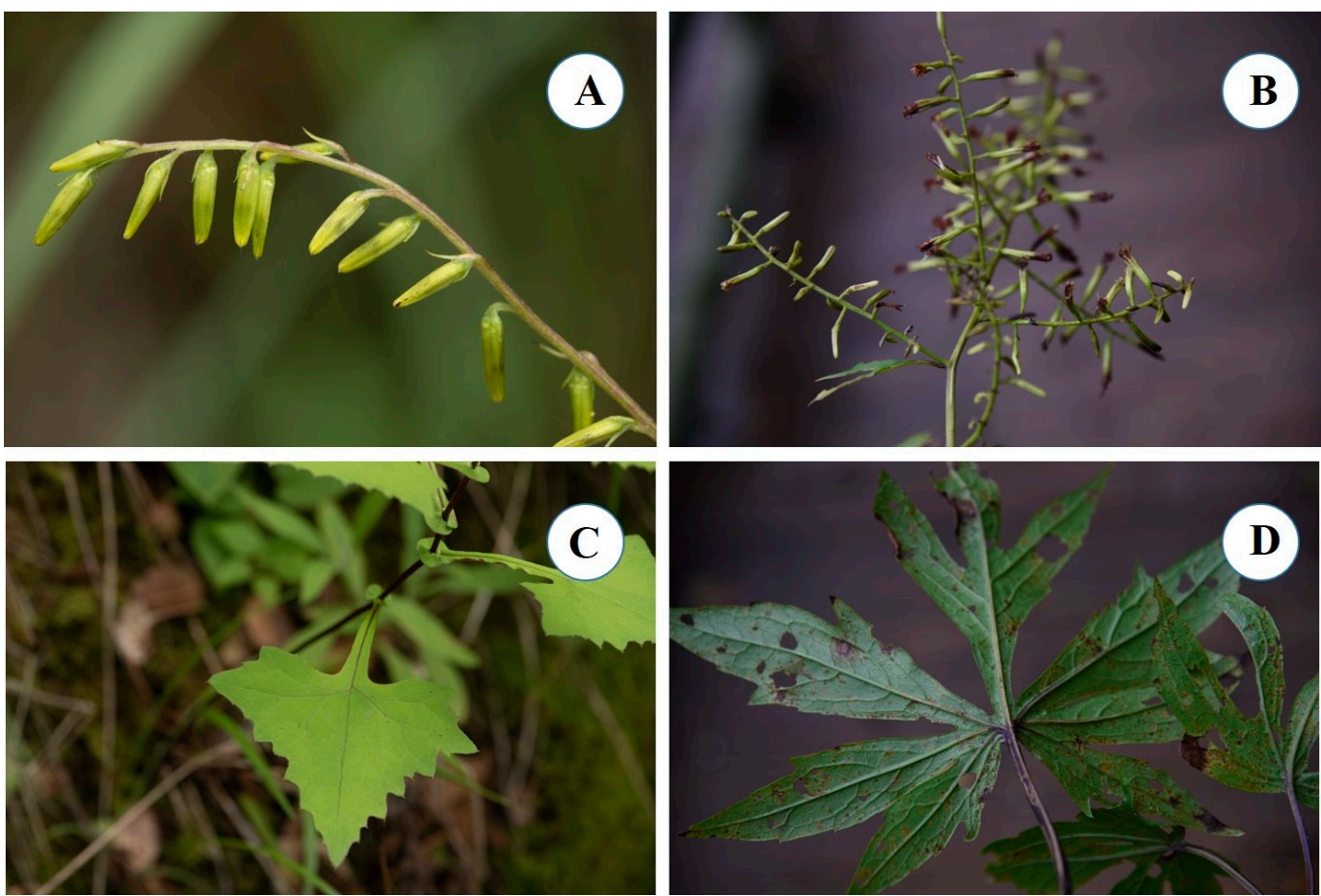

**Figure 1.** Images of *Parasenecio latipes* and *P. palmatisectus*. (**A**,**B**) Inflorescence of plant growing in natural habitat; (**C**,**D**) morphology of leaf. ((**A**,**C**) *P. latipes*, voucher FZX5888, SCNU, Muli county, Sichuan province, China). ((**B**,**D**) *P. palmatisectus*, voucher DY122, SCNU, Guangwu mountain, Bazhong city, Sichuan province, China). Photographs by Zhixi Fu.

*2.3. Repeat Sequence Analysis*

Interspersed repeat sequences analysis include forward, reverse, complement, and palindrome repeats. The REPuter program (https://bibiserv.cebitec.unibielefeld.de/reputer/ (accessed on 15 June 2022)) was used to identify the interspersed repeat sequences in the two cp genomes and the related eight species of the tribe Senecioneae [42]. The parameters used in the analysis were as follows: hamming distance = 3, maximum computed repeats size = 50 bp, and minimal repeat size = 30 bp. The SSRs in the plastomes were detected by Perl script MISA [43]. The repeat units were set to 10 for mononucleotides, 5 for dinucleotides, 4 for trinucleotides, and 3 for hexanucleotides.

*2.4. Comparative Genome Analysis and Molecular Marker Identification*

In order to observe the extent of the difference in the *Parasenecio* species and the eight species of LSC/SSC/IR, we used published cp genome sequences of eight species of Senecioneae. They were *Ligularia fischeri* Turcz. (GenBank accession number MG729822), *L. hodgsonii* Hook.f. (no. MF539929), *L. intermedia* Nakai (no. MF539930), *L. jaluensis* Kom. (no. MF539931), *L. mongolica* DC. (no. MF539932), *L. veitchiana* Greenm. (no. MF539933), *Sinosenecio baojingensis* Ying Liu & Q.E.Yang (no. MZ325394), and *S. jishouensis* D.G. Zhang, Ying Liu & Q. E. Yang (no. MT876597). Full alignment with annotation was visualized using the online genome comparison tool mVISTA (http://genome.lbl.gov/vista/mvista/submit.shtml (accessed on 15 June 2022)) [44]. The boundaries of four regions in the cp genomes were compared using Irscope (https://irscope.shinyapps.io/irapp/ (accessed

on 15 June 2022)) [45]. To identify nucleotide variability pi (π), the 10 sequences of the cp genome were aligned using MAFFT v7.475 [46]. The nucleotide diversity of the cp genomes was calculated based on sliding window analysis using the DnaSP v5.10 software (http://www.ub.edu/dnasp/ (accessed on 6 January 2022)) [47]. The window size was set to 800 bp with a step size of 200 bp.

### 2.5. Phylogenetic Analysis

To reconstruct the phylogenetic tree, 46 cp genome sequences were used for the phylogenetic analysis, including 2 newly sequenced *Parasenecio* species, 42 taxa of Asteraceae, and 2 outgroup taxa of *Anthriscus cerefolium* (Apiaceae) and *Kalopanax septemlobus* (Araliaceae). Two data matrices (complete cp genome and intergenic spacer (IGS) sequence) were selected for phylogenetic analysis. Sequence alignment was achieved using the MAFFT 7.475 with default parameters [46]. A maximum likelihood (ML) analysis was carried out with RaxML v7.2.8 based on the GTRGAMMA model on the CIPRES (https://www.phylo.org/ (accessed on 20 June 2022)) using 1000 bootstrap replicates [48,49].

## 3. Results

### 3.1. Features of the CP Genomes

The cp genomes of the two *Parasenecio* species were very conserved. The complete cp genomes of *P. latipes* and *P. palmatisectus* obtained in this study were 151,185 bp and 151,263 bp in length, respectively (Figure 2 and Table 1). The cp genomes of *P. latipes* and *P. palmatisectus* contained an SSC region (18,221 bp and 18,251 bp), an LSC regions (83,308 bp and 83,352 bp), and two inverted repeats (24,828 bp and 24,830 bp), respectively (Table 1). The total GC content was 37.5% in the complete cp genome, 43.0% in IR, 35.6% in LSC, and 30.7–30.8% in SSC. All plastomes possessed 133 genes, including 86 protein-coding genes, 37 transfer RNA (tRNA) genes, 8 ribosomal RNA (rRNA) genes, and 2 pseudogenes (*ycf1* and *rps19*) (Table 2). The gene composition of *Parasenecio* species could be divided into four categories: photosynthesis-related, self-replication-related genes, protein-coding genes with unknown functions, and other genes. Seven protein-coding genes (*ndhB*, *rpl2*, *rpl23*, *rps12*, *rps7*, *ycf15*, and *ycf2*), seven tRNA genes (*trnA-UGC*, *trnI-CAU*, *trnI-GAU*, *trnL-CAA*, *trnN-GUU*, *trnR-ACG*, and *trnV-GAC*), and all rRNA genes (*4.5S*, *5S*, *16S*, and *23S*) were located within the IR regions (Table 2). Ten of the protein-coding genes and six of the tRNA genes contained introns, thirteen of which contained a single intron, and three genes (*rps12*, *ycf3*, *clpP*) had two introns (Table 2). In particular, *rps12* is a trans-spliced gene, with the first exon residing in the LSC region, and the second and third exons residing in the IR regions (Figure 2).

**Table 1.** Comparative analyses of cp genomes among 10 species of Senecioneae.

| Species | GenBank No. | Genome Size (bp) | LSC (bp) | IR (bp) | SSC (bp) | GC Content (%) | | | | Number of Functional Genes | | | |
|---------|-------------|------------------|----------|---------|----------|------|------|------|------|-------|-----|-------|-------|
| | | | | | | All | LSC | IR | SSC | Total | CDS | rRNAs | tRNAs |
| *L. fischeri* | MG729822 | 151,193 | 83,301 | 24,830 | 18,232 | 37.5 | 35.6 | 43.0 | 30.7 | 132 | 87 | 8 | 37 |
| *L. hodgsonii* | MF539929 | 151,136 | 83,253 | 24,833 | 18,217 | 37.5 | 35.6 | 43.0 | 30.7 | 132 | 87 | 8 | 37 |
| *L. intermedia* | MF539930 | 151,152 | 83,258 | 24,831 | 18,232 | 37.5 | 35.6 | 43.0 | 30.7 | 132 | 87 | 8 | 37 |
| *L. jaluensis* | MF539931 | 151,148 | 83,263 | 24,830 | 18,225 | 37.5 | 35.6 | 43.0 | 30.7 | 132 | 87 | 8 | 37 |
| *L. mongolica* | MF539932 | 151,118 | 83,244 | 24,830 | 18,214 | 37.5 | 35.6 | 43.0 | 30.7 | 132 | 87 | 8 | 37 |
| *L. veitchiana* | MF539933 | 151,253 | 83,330 | 24,838 | 18,247 | 37.5 | 35.6 | 43.0 | 30.7 | 132 | 87 | 8 | 37 |
| *P. latipes* | ON749759 | 151,185 | 83,308 | 24,828 | 18,221 | 37.5 | 35.6 | 43.0 | 30.8 | 131 | 86 | 8 | 37 |
| *P. palmatisectus* | ON749760 | 151,263 | 83,352 | 24,830 | 18,251 | 37.5 | 35.6 | 43.0 | 30.7 | 131 | 86 | 8 | 37 |
| *S. baojingensis* | MZ325394 | 151,315 | 83,445 | 24,849 | 18,172 | 37.4 | 35.5 | 43.0 | 30.6 | 132 | 87 | 8 | 37 |
| *S. jishouensis* | MT876597 | 151,257 | 83,373 | 24,853 | 18,178 | 37.4 | 35.5 | 43.0 | 30.6 | 134 | 89 | 8 | 37 |

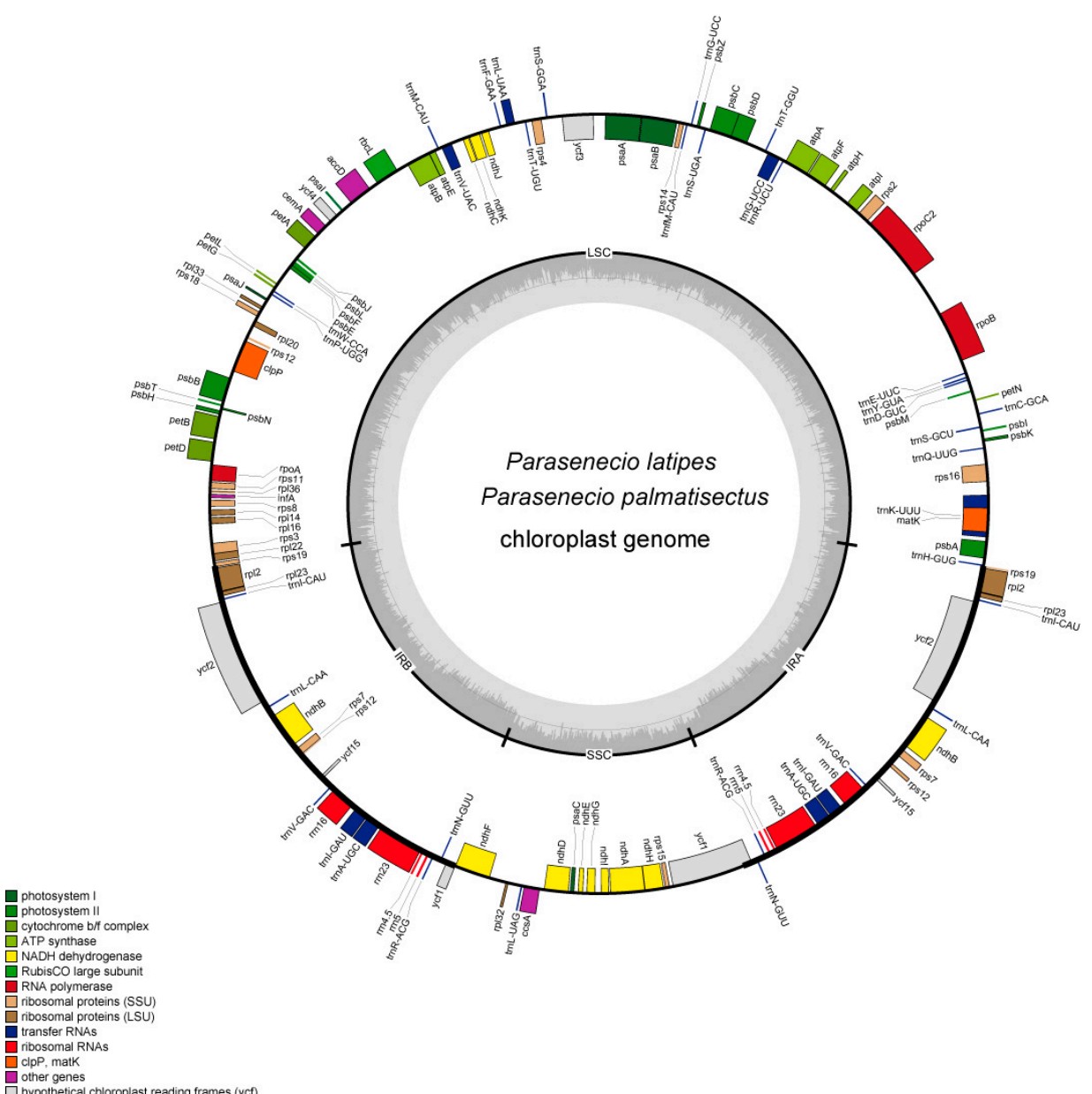

**Figure 2.** The cp genome maps of *P. latipes* and *P. palmatisectus.* Genes shown on the outside of the circle are transcribed clockwise, and genes inside are transcribed counter-clockwise. Genes are color-coded to indicate functional group. The darker gray inner circle corresponds to the GC content, whereas the lighter gray indicates the AT content.

The cp genomes of eight Senecioneae species were selected for comparison with the two *Parasenecio* species (Table 1). As expected, all the plastomes of the 10 species showed a typical quadripartite structure. The plastomes of the 10 Senecioneae species were 151,118 to 151,315 bp, with little variation in length between the newly generated and published genomes. The total GC contents of the plastomes were highly similar (37.4 to 37.5%), while the different regions (LSC, SSC, and IR) had slightly variable GC contents. There was little difference in total genes (131–134) and CDS (86–89) between *Parasenecio* and the other genera of Senecioneae. The *S. jishouensis* species has the most genes and two *Parasenecio* species have the least genes.

**Table 2.** List of genes found in the two cp genomes of *Parasenecio* species.

| Category of Genes | Group of Genes | Name of Genes | Number of Genes |
|---|---|---|---|
| Photosynthesis | Subunits of photosystem I | *psaA, psaB, psaC, psaI, psaJ* | 5 |
| | Subunits of photosystem II | *psbA, psbB, psbC, psbD, psbE, psbF, psbH, psbI, psbJ, psbK, psbL, psbM, psbN, psbT, psbZ* | 15 |
| | Subunits of NADH dehydrogenase | *ndhA \*, ndhB \* (×2), ndhC, ndhD, ndhE, ndhF, ndhG, ndhH, ndhI, ndhJ, ndhK* | 12 |
| | Subunits of cytochrome b/f complex | *petA, petB \*, petD \*, petG, petL, petN* | 6 |
| | Subunits of ATP synthase | *atpA, atpB, atpE, atpF \*, atpH, atpI* | 6 |
| | Large subunit of rubisco | *rbcL* | 1 |
| Self-replication | Proteins of large ribosomal subunit | *rpl14, rpl16, rpl2 \* (×2), rpl20, rpl22, rpl23(×2), rpl32, rpl33, rpl36* | 11 |
| | Proteins of small ribosomal subunit | # *rps19, rps11, rps12 \*\* (×2), rps14, rps15, rps16 \*, rps18, rps19, rps2, rps3, rps4, rps7(×2), rps8* | 15 |
| | Subunits of RNA polymerase | *rpoA, rpoB, rpoC2* | 3 |
| | Ribosomal RNAs | *rrn16(×2), rrn23(×2), rrn4.5(×2), rrn5(×2)* | 8 |
| | Transfer RNAs | *trnA-UGC \* (×2), trnC-GCA, trnD-GUC, trnE-UUC, trnF-GAA, trnG-UCC, trnG-UCC \*, trnH-GUG, trnI-CAU(×2), trnI-GAU \* (×2), trnK-UUU \*, trnL-CAA(×2), trnL-UAA \*, trnL-UAG, trnM-CAU, trnN-GUU(×2), trnP-UGG, trnQ-UUG, trnR-ACG(×2), trnR-UCU, trnS-GCU, trnS-GGA, trnS-UGA, trnT-GGU, trnT-UGU, trnV-GAC(×2), trnV-UAC \*, trnW-CCA, trnY-GUA, trnfM-CAU* | 37 |
| Other genes | Maturase | *matK* | 1 |
| | Protease | *clpP \*\** | 1 |
| | Envelope membrane protein | *cemA* | 1 |
| | Acetyl-CoA carboxylase | *accD* | 1 |
| | c-type cytochrome synthesis gene | *ccsA* | 1 |
| | Translation initiation factor | *infA* | 1 |
| Genes of unknown function | Conserved hypothetical chloroplast ORF | # *ycf1, ycf1, ycf15(×2), ycf2(×2), ycf3 \*\*, ycf4* | 8 |

Notes: \* Gene contains one intron. \*\* Gene contains two introns. #: Pseudogene. (×2) indicates that the number of the repeat unit is 2.

*3.2. Repeat Sequences Analysis*

Repeat sequences included SSRs and interspersed repeat sequences. In total, 51 SSRs were identified in *P. latipes* and 49 in *P. palmatisectus*, which is similar to other closely related taxa. Furthermore, the number of SSRs varied in the 10 species and ranged from 49 (*P. palmatisectus* and *S. baojingensis*) to 61 (*L. veitchiana*). Among the mono-, di-, tri-, tetra-, penta-, and hexanucleotide categories of SSRs in the cp genomes of the Senecioneae, mononucleotide repeats were the most abundant, pentanucleotide repeats were only present in *S. jishouensis*, but no hexanucleotide was found in the 10 Senecioneae species (Figure 3). There were five types of dominant motifs in SSRs: A/T, AT/TA, AAT/ATT, AAAT/ATTT, and AATAT/ATATT (Figure 3).

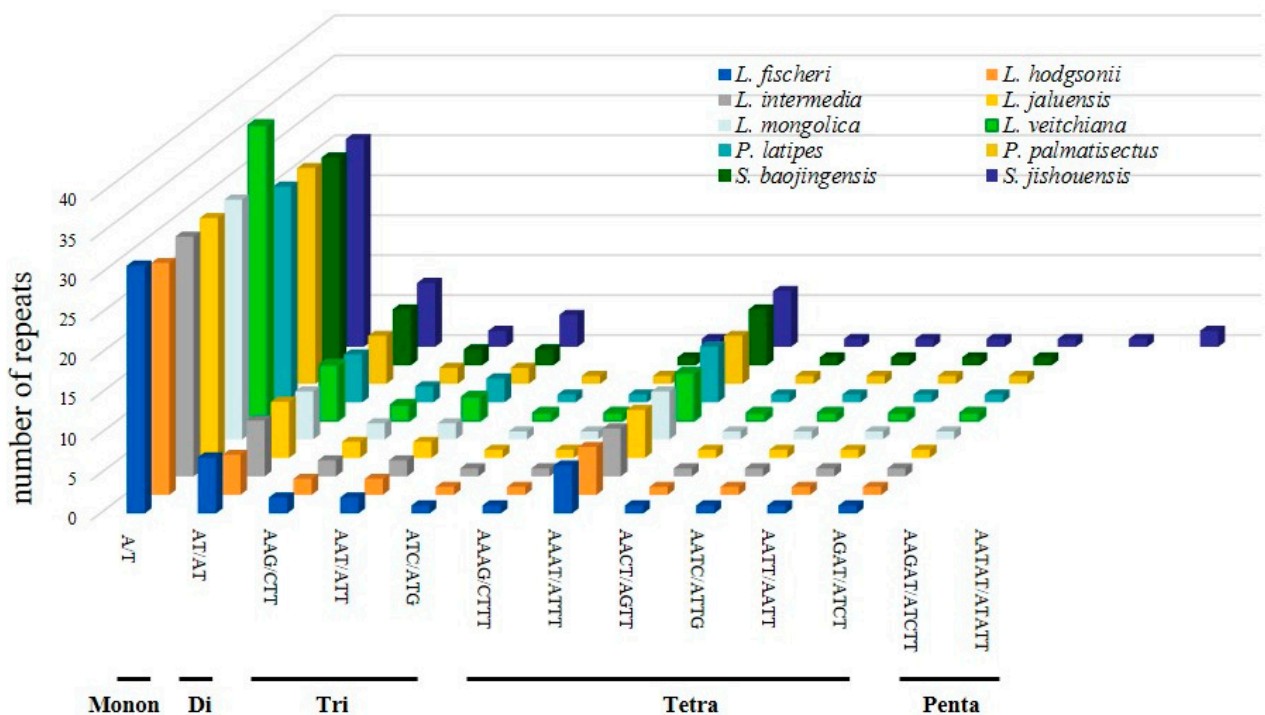

**Figure 3.** Number of identified SSR motifs in different repeat class types of 10 Senecioneae species of cp genomes.

Overall, 34 interspersed repeats were identified in *P. latipes* and 35 in *P. palmatisectus*. A total of 389 (34 to 46 in each species) interspersed repeat sequences, including 185 forward repeats, 200 palindromic repeats, 2 reverse repeats, and 2 complement repeats, were identified in the plasetomes of 10 Senecioneae species. Forward repeats (16 to 21 in each species) and palindrome repeats (18 to 23 in each species) were found in all species (Figure 4A). Among these repeats, most of the repeat units were composed of 30 to 34 bp, followed by repeats of 35 to 39 bp, 40 to 44 bp, and 45 to 49 bp, while repeat units > 55 bp were comparatively rare (Figure 4B).

*3.3. IR Contraction and Expansion*

The comparison of LSC/IRb/SSC/IRa boundary regions in the plastomes from 10 Senecioneae species (*L. Fischeri*, *L. hodgsonii*, *L.intermedia*, *L. jaluensis*, *L. mongolica*, *L. veitchiana*, *S. baojingensis*, and *S. jishouensis*) is presented in Figure 5. There was little difference among the Senecioneae plastomes. The genes *rps19*, *rpl2*, *ndhF*, *ycf1*, *trnN*, and *trnH* were located at the boundary regions. The *rps19* gene crossed over the LSC/IRb boundary and extended into the IRb region, which ranged from 60 bp to 62 bp. The *rps19* pseudogene existed in all cp genomes at the IRa/LSC boundary except for *S. baojingensis*. In all species, the *rpl2* gene was included within the IRb region in all species, 113 to 116 bp away from the LSC/IRb boundary. The *ndhF* gene was entirely present within the SSC region in the Senecioneae species except *S. baojingensis* and *S. jishouensis*, where the *ndhF* gene extended into the IRb region with 1 bp. All 10 Senecioneae species contained a functional copy of the *ycf1* gene at the SSC/IRa boundary and its pseudocopy (*ycf1*Ψ) at the IRb/SSC boundary. In addition, *ycf1*Ψ ranged in size from 578 bp to 597 bp. The *trnN* gene was solely located in the IRa region, 905 to 921 bp away from the SSC/IRa boundary. The *trnH* gene was located at the IRa/LSC boundary, and it was totally contained in the LSC region.

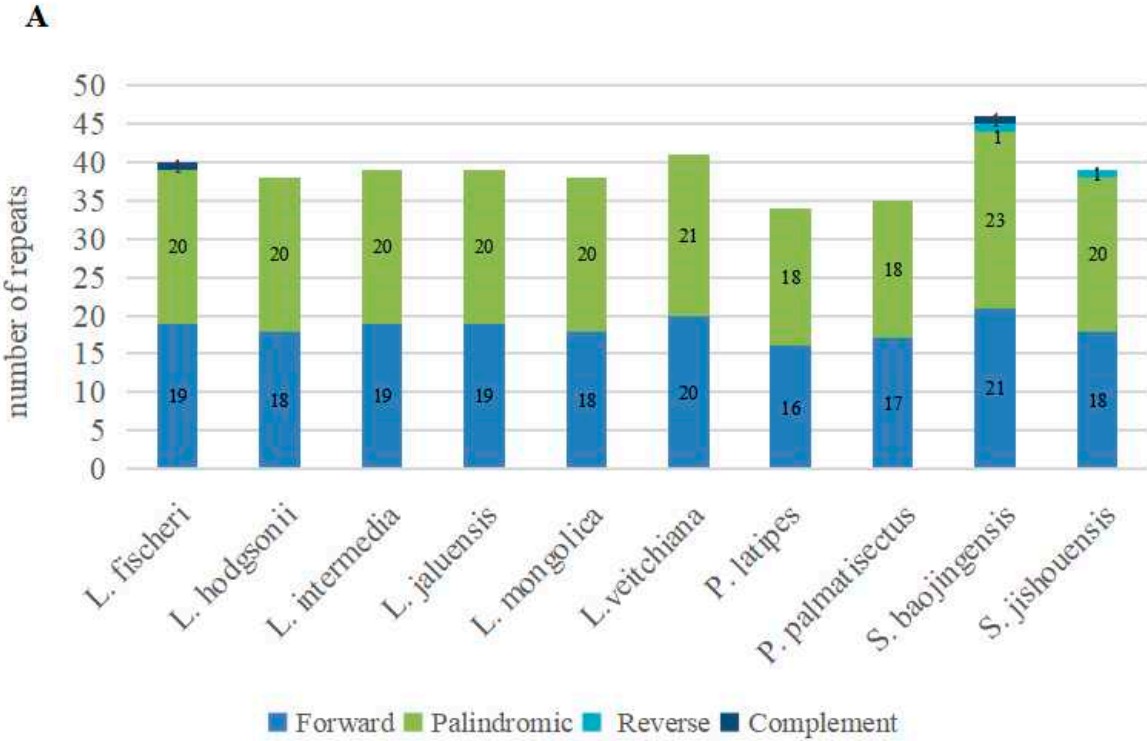

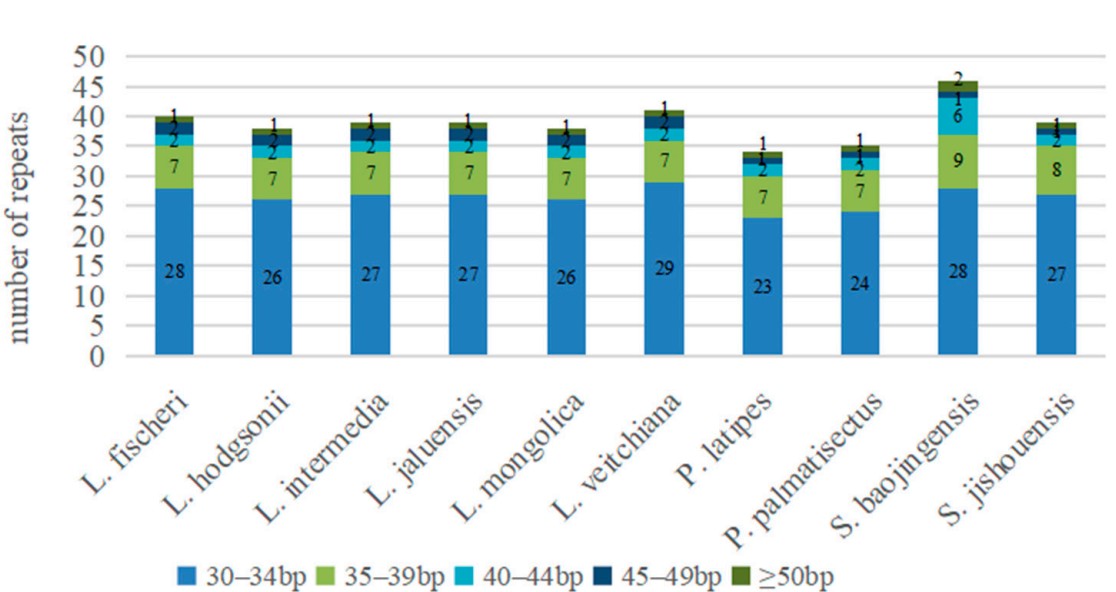

**Figure 4.** Analysis of interspersed repeat sequences in the cp genomes from 10 Senecioneae species. (**A**) The quantity of four types of repeats in different species. (**B**) The number of repeats with different lengths in different species.

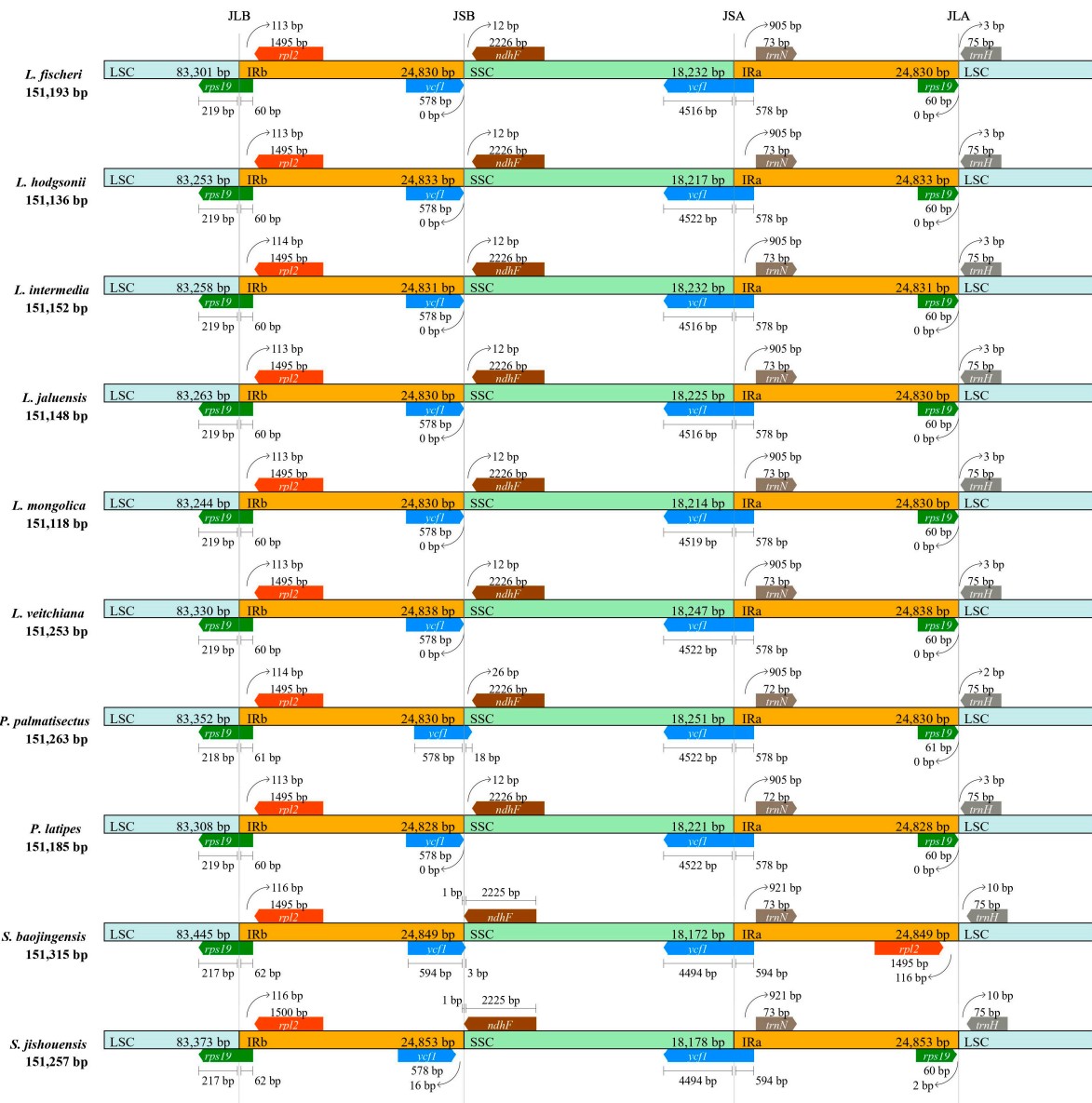

**Figure 5.** Comparison of the border positions of LSC, SSC, and IR regions among cp genomes from 10 Senecioneae species. Gene names are indicated in boxes.

### 3.4. Sequence Divergence Analysis

The structural differences among Senecioneae plastomes were compared by mVISTA (Figure 6). Overall, the LSC and SSC regions in these cp genomes were more divergent than the IR regions. Noncoding regions exhibited a higher divergence than coding regions. A sliding window analysis revealed highly variable regions in the Senecioneae plastomes (Figure 7). The nucleotide diversity (PI) value ranged from 0 to 0.01189. The variability in the IR regions was lower than in the LSC and SSC regions, which was consistent with the mVISTA results. There were six highly variable hotspots that showed significantly higher PI values (>0.009), including two gene regions (*ycf1* and *ccsA*) and four intergenic regions (*trnC-GCA-petN*, *ycf1-trnN-GUU*, *psaI-ycf4*, and *rpl32-trnL-UAG*). These hot regions could be used as potential molecular markers for phylogenetic studies of *Parasenecio* species.

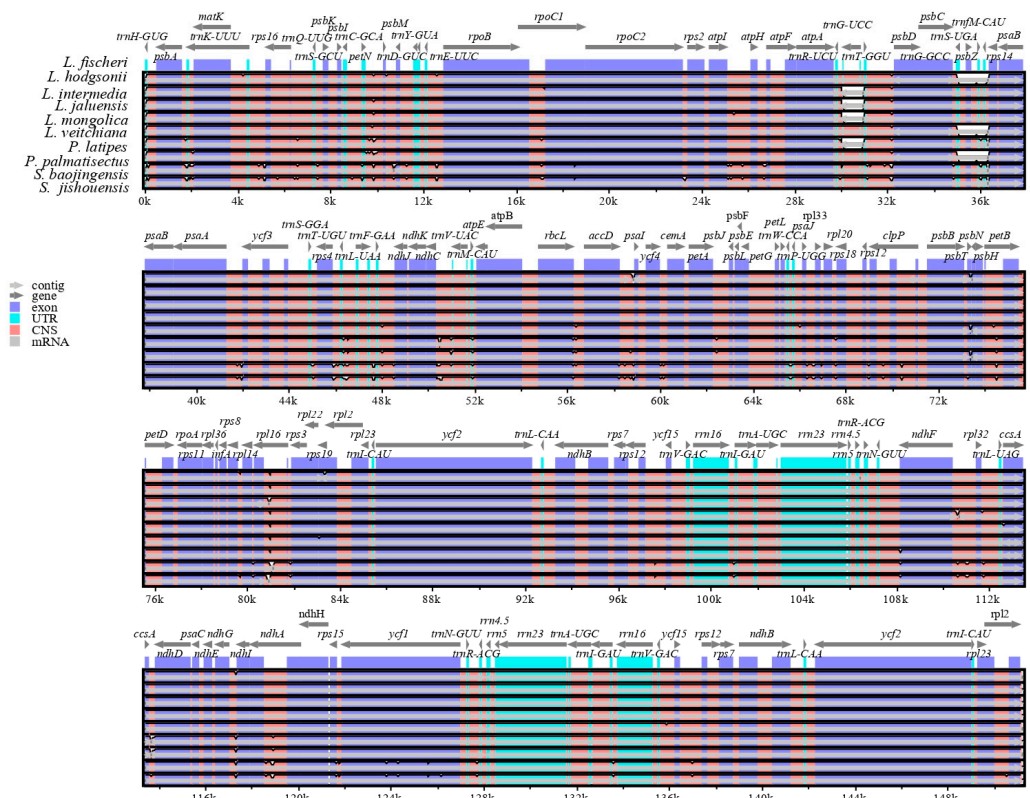

**Figure 6.** The homologous comparison of 10 Senecioneae plastomes by mVISTA with *L. fischeri* as a reference. Gray arrows and thick black lines above the alignment indicate genes with their orientation and the position of the IRs, respectively. A cut-off value of 70% identity was used for the plots, and the Y-scale represents a percent identity between 50% and 100%.

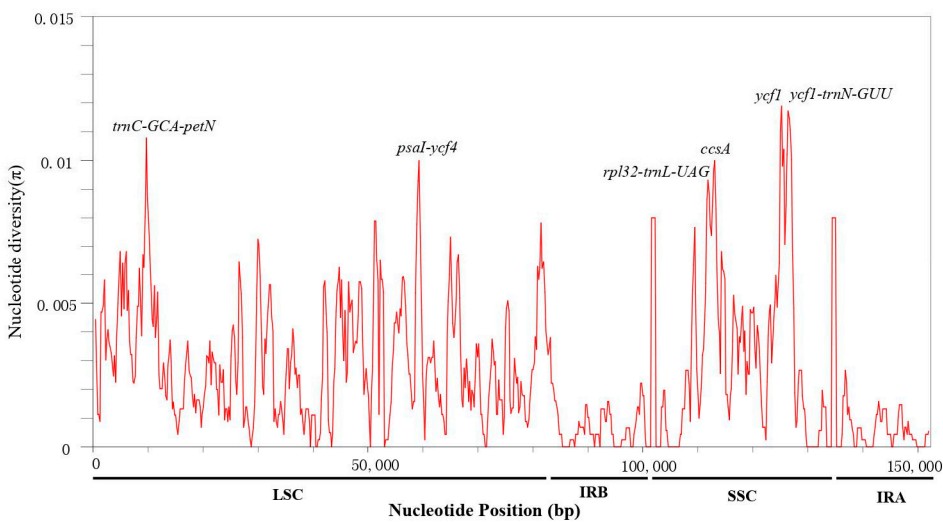

**Figure 7.** Sliding window analysis of the whole chloroplast genomes of 10 species of tribe Senecioneae. Window length: 800 bp; step size: 200 bp; *X*-axis: position of the midpoint of a window; *Y*-axis: nucleotide diversity of each window.

### 3.5. Phylogenomic Analysis

Two cp genomes were sequenced in this study, forty-two representative Asteraceae species were chosen to construct the phylogenetic tree, and *A. cerefolium* (Family Apiaceae) and *K. septemlobus* (Family Araliaceae) were included as outgroups (Figure 8). The phylogenetic relationships based on complete cp genomes and IGS sequence were consistent. The

phylogenetic analysis showed that the genus *Parasenecio* is sister to the genera of *Ligularia* and *Sinosenecio*. *Parasenecio latipes* and *P. palmatisectus* are located in the Senecioneae tribe of Asteroideae. The evolutionary tree revealed clear phylogenetic relationships for 42 species in 10 tribes of Asteraceae, which were clustered into 5 clades. The first clade consists of 37 species in 6 tribes: Senecioneae, Anthemideae, Gnaphalieae, Astereae, Heliantheae, and Inuleae, all belonging to Asteroideae. The second clade consists of three species from tribe Cichorieae, belonging to Cichorioideae. The third clade consists of one species, belonging to Gymnarrheneae. The fourth clade consists of two species, belonging to Carduoideae. The fifth clade consists of one species, belonging to Pertyoideae.

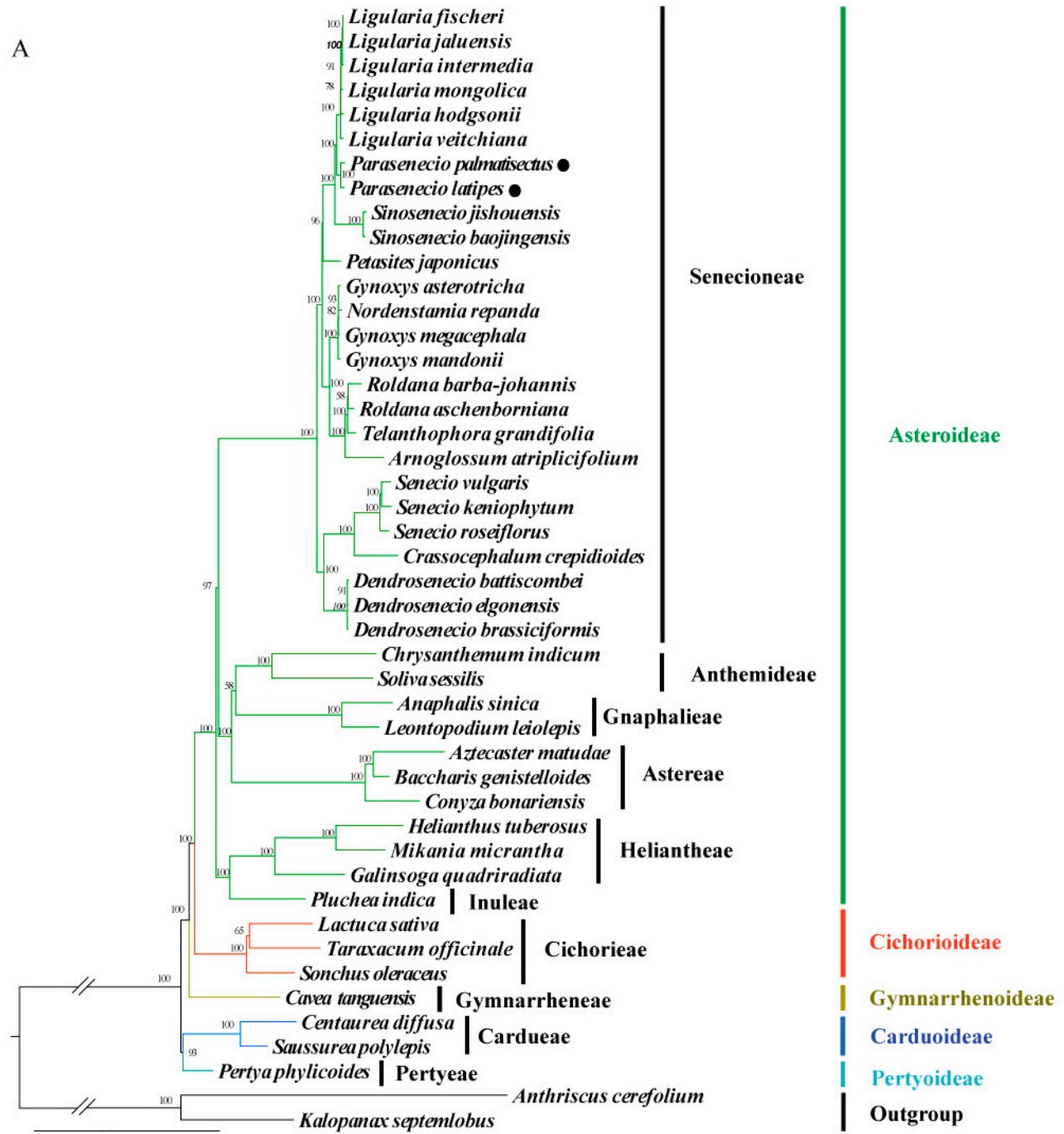

**Figure 8.** *Cont.*

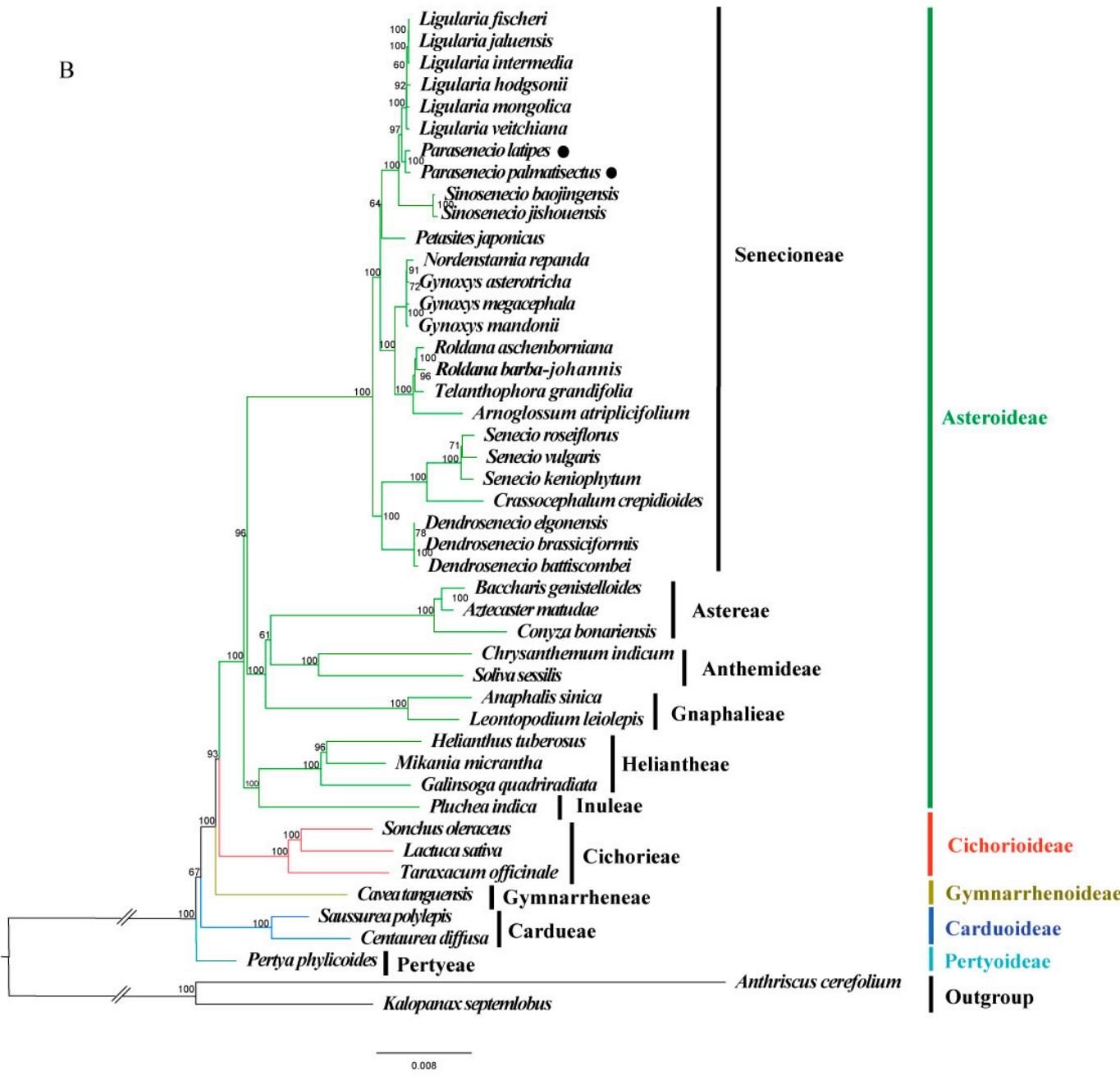

**Figure 8.** Phylogenetic tree reconstruction of 46 taxa using ML method based on (**A**) complete cp genomes and (**B**) the IGS sequence. Bootstrap support values are shown at the nodes. The phylogenetic position of *Parasenecio* species is marked with black circles.

## 4. Discussion

In this study, the complete cp genomes of two *Parasenecio* species were sequenced, and comparative analyses were carried out with the other eight Senecioneae species to detect the differences in the cp genome. The results showed that both *Parasenecio* species had typical quadripartite structures (Figure 2). Our findings are consistent with the cp genome reports of other studies of Senecioneae [22,27]. The two *Parasenecio* species exhibited similar numbers of protein-coding genes, tRNAs, and rRNAs. The average GC content of the complete cp genomes in the two species was 37.5% (Table 1), which was similar to the findings of previous studies [22,25,27,28]. Moreover, the GC contents among the cp genomes of the 10 Senecioneae species were similar, and the IR regions (43.0%) had higher

GC contents than the single-copy regions (LSC: 35.6% and SSC: 30.6% to 30.8%) (Table 1). This result is consistent with previous reports [25–28]. The high GC contents in the IR regions could be due to the presence of four rRNA genes (*rrn16*, *rrn23*, *rrn4.5*, and *rrn5*).

The structure and genes of the cp genome of *Parasenecio* species were highly conserved. Nevertheless, there was a difference in the sizes of cp genomes. This phenomenon may be caused by contractions and expansions in the boundary regions [20]. The *ψycf1* and *ψrps19* pseudogenes were present in *Parasenecio* species. A previous study also reported pseudogenes in the cp genome of Asteraceae species [17,22].

Due to their high mutation rate, the SSR markers are frequently used for genetic evolutionary and species identification analyses [50–53]. In this study, appropriately, 49 to 61 SSRs were identified in the cp genome of Senecioneae species. The A/T mononucleotides were the most abundant SSRs (Figure 3), implying that there were more replications in Senecioneae species. Similar patterns of SSRs distribution were also reported in the cp genomes of other Asteraceae genera or tribes [27,33]. The newly detected SSRs identified in this study will be useful for the development of effective molecular markers for the *Parasenecio* species in future studies.

In previous molecular studies, a large number of highly variable regions were used as DNA barcodes for species identification [20,23,35]. In this study, six highly variable regions (*trnC-GCA-petN*, *ycf1-trnN-GUU*, *psaI-ycf4*, *rpl32-trnL-UAG*, *ycf1*, and *ccsA*) were identified in the tribe Senecioneae based on a comparative analysis of the cp genome (Figure 7). These highly variable regions could be useful for further studies concerning species identification and the phylogenetic relationships of *Parasenecio* species. Among these regions, *ycf1* and *psaI-ycf4* were demonstrated to be conducive markers for phylogenetic studies [54–57].

These similar cp genomes provide an ideal resource for phylogenetic studies [58,59]. To date, several studies have been conducted to establish and analyze phylogenetic relationships in the Asteraceae family. The cp gene has been used successfully to infer phylogenetic relationships within Anthemideae [20], Senecioneae [22,27], and Gymnarrheneae [35]. However, the phylogenetic position of the *Parasenecio* species is still lacking. In this study, the complete cp genome sequence data from 13 genera of Senecioneae were used to construct the phylogenetic relationship of *Parasenecio* and its related genera. The phylogenetic analysis showed that the genus *Parasenecio* is sister to the genera of *Ligularia* and *Sinosenecio* (Figure 8). The phylogenetic relationships identified among *Parasenecio* species were consistent with those from previous studies [7–11]. For example, Liu et al. named the clade the *Ligularia–Cremanthodium–Parasenecio* (LCP) complex [7]. The relationships within this clade have been studied using the nuclear internal transcribed spacer (ITS) [8]. This result was congruent with previous research [9–11]. Our results may be more reliable, since we used the complete cp genome and the IGS sequence in phylogeny reconstruction. In *Parasenecio*, previous studies only used a few cp genomes [12] and nrDNA ITSs [7]. However, the cp genomes of *Parasenecio* were not sequenced. In this study, the comparative cp genome data provide efficient molecular markers. Our results revealed that the divergence hotspot regions were *trnC-GCA-petN*, *ycf1-trnN-GUU*, *psaI-ycf4*, *rpl32-trnL-UAG*, *ycf1*, and *ccsA* in *Parasenecio* species. These highly variable regions could be used as molecular markers for future phylogenetic studies of *Parasenecio* species. Our results provide valuable genomic information for further studies on the phylogenetic relationships and sustainable utilization of *Parasenecio* species.

## 5. Conclusions

In this study, we assembled, annotated, and analyzed the cp genomes of *P. palmatisectus* and *P. latipes*. The complete cp genome sizes of *P. latipes* and *P. palmatisectus* were 151,185, and 151,263 bp, respectively. Both cp genomes contained 133 genes, including 86 protein-coding genes, 37 tRNA genes, 8 rRNA genes, and 2 pseudogenes (*ycf1* and *rps19*). A comparative genomic analysis showed that the complete cp genome sequences of *Parasenecio* species and their related species were relatively conserved. A total of six highly variable regions were identified, including two gene regions (*ycf1* and *ccsA*) and

four intergenic regions (*trnC-GCA-petN*, *ycf1-trnN-GUU*, *psaI-ycf4*, and *rpl32-trnL-UAG*). These could be used as potential markers for further phylogenetic and population genetic studies of the tribe Senecioneae. A phylogenetic analysis based on the complete cp genome and IGS sequence showed that the genus *Parasenecio* is a sister genus to *Ligularia* and *Sinosenecio*. *Parasenecio latipes* and *P. palmatisectus* are located in the Senecioneae tribe of Asteroideae. The cp genomic resources presented in this study provide information for studies on genetic diversity, species identification, and phylogenetics in *Parasenecio* species and other closely related species.

**Author Contributions:** D.W. and Z.F. conceived and designed the research. M.Z. and X.L. performed bioinformatic analyses. X.L., Q.W. and J.L. (Junjia Luo) carried out wet-lab experiments. J.L. (Jie Liu) contributed to data interpretation. X.L., D.W. and Z.F. wrote the manuscript. D.W. and Z.F. revised the manuscript. All authors have read and agreed to the published version of the manuscript.

**Funding:** This study was financially supported by the National Natural Science Foundation of China (No. 32000158, 62002250), the National Science & Technology Fundamental Resources Investigation Program of China (No. 2021XJKK0702) and the Foundation of Sustainable Development Research Center of Resources and Environment of Western Sichuan, Sichuan Normal University (No. 2020CXZYHJZX03).

**Institutional Review Board Statement:** Not applicable.

**Data Availability Statement:** Not applicable.

**Acknowledgments:** The authors would like to thank the editor and anonymous reviewers for the constructive criticism of the original manuscript.

**Conflicts of Interest:** The authors declare no conflict of interest.

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
