# Peer review of "Phylogenomic Analysis of Two Species of Parasenecio and Comparative Analysis within Tribe Senecioneae (Asteraceae)"

_diversity, doi:10.3390/d15040563_

Round 1
Reviewer 1 Report
This article present Phylogenomic Analysis of Two Species of Parasenecio and Comparative Analysis Within Tribe Senecioneae (Asteraceae). Before recommending this article for publication, there are some shortcomings for that should be resolve.
The work is well presented but some corrections are required.
In abstract the results should be provided in sequence wise.
Quantitative results must be included in the abstract.
Which methods or techniques were used must be mentioned in the abstract.
Add economic and medicinal importance of the genus, its origin, species distribution.
Also add specifically economic and medicinal importance of the studied samples, its origin and distribution.
Line 45 should be cited with more recent studies. The following studies could be helpful.
https://doi.org/10.3390/ijms22179175, Pak. J. Bot., 54(3): DOI: http://dx.doi.org/10.30848/PJB2022-3(19)
Why only these two species were selected in this study. Make a problem statement at the end of the introduction.
Add a paragraph on the significance and novel techniques of the phylogenomic.
Web links should be provided in the methods with respective software.
Add figures in the respective sections according to the journal format. Please read author guidelines.
Reviewer 2 Report
This manuscript sequence two Parasenecio chloroplast genome and performed comprehensive analyses combining other 44 species from the genome structure, gene numbers, SSR, and phylogenetic relationship. My suggestion is: major revision.
Several major points that the authors need to be addressed as follows:
1. the authors focus on the two genus Parasenecio species, and made various comparative analyses, however, no species pictures were shown in the text, it is better to show the morphological traits of these species, such as plant, flower and leaf, which may be more visual for readers.
2. approximately 70 species in Parasenecio, however, only two Parasenecio species chloroplast genomes were collected in this study, it is meanless to discuss the monophyly on this genus.
3. Whole cp genome sequences were used to conduct the phylogenetic analysis, it may be inappropriate. Indeed, some genes or regions are missing in part of Senecioneae species, which may cause huge sequence differences and lead to incorrect phylogenetic results. It is better to extract the Genes, CDS, and Noncoding sequences, respectively, and separately use the three sequence-data sets to carry a concatenated phylogenetic analysis. (you can see the following two publications:
1. Xie DF, Tan JB, Yu Y, et al., Insights into phylogeny, age and evolution of Allium (Amaryllidaceae) based on the whole plastome sequences. Ann Bot. 2020 1;125(7):1039-1055.
2. Xie DF, Yu HX, Price M, et al. Phylogeny of Chinese Allium Species in Section Daghestanica and Adaptive Evolution of Allium (Amaryllidaceae, Allioideae) Species Revealed by the Chloroplast Complete Genome. Frontiers in plant science, 2019, 10, 460. )
4. in view of the less Parasenecio chloroplast genomes used in this study, if there are some difficulties in materials collections, the authors should download more cp genes regions (such as psbK, trnL-trnF, ndhF) to perform a phylogenetic analysis, I can see there are many cp genes sequences can be obtained from the NCBI. therefore, a comprehensive phylogeny of genus Parasenecio can be detected.
5. if some nucleotide sequences can be collected from NCBI, for example ITS (internal transcribed spacers), it is better to make a comparative analysis between cp genes and nucleotide genes.

Reviewer 3 Report
Overall, the manuscript describes a significant phylogenetic analysis under maximum likelihood that showed the two Parasenecio species formed a monophyletic group which was sister to the genera of Ligularia and Sinosenecio in the tribe Senecioneae. Although, manuscript is supported with sound scientific evidence, it needs careful revision with following points:
First, author need to revise some of the recent literature on the molecular markers for crop improvement. I recommend some SSR/RAPD applications that could be included to enrich introduction and discussion section of this manuscript. For example,
Multiplex molecular marker-assisted analysis of significant pathogens of cotton (Gossypium sp.), 2022; Biocatalysis and Agricultural Biotechnology https://doi.org/10.1016/j.bcab.2022.102557 (Cotton); Assessment of genetic diversity and volatile content of commercially grown banana (Musa spp.) cultivars, Hinge et al., Scientific Reports, 2022; https://doi.org/10.1038/s41598-022-11992-1 (Banana); Microsatellite and RAPD analysis of grape (Vitis spp.) accessions and identification of duplicates/misnomers in germplasm collection, Upadhyay et al., 2010 Indian J Hortic Volume 67 Pages 8-15; Microsatellite analysis to differentiate clones of Thompson seedless grapevine, Upadhyay et al., 2010, Ind Journal of Horticulture, Volume 67 Issue 2 Pages 260-263.
Figure 2. Please elaborate the caption for this figure, it too short and not easy to follow.
On page 10, two figures are pasted, however, one figure’s caption missing. Please rectify this, by combining A and B on same page.
For Figure 7. Describe the phylogenetic tree in caption as well.
Additionally, provide key outputs of genetic analysis and signify the key implications of molecular markers.
Round 2
Reviewer 2 Report
I can see the authors have paid careful attention to the comments and suggestions. The improvement in the writing is noticeable and most welcome. the present version can be accepted.
Author Response
Reviewer 2:
I can see the authors have paid careful attention to the comments and suggestions. The improvement in the writing is noticeable and most welcome. the present version can be accepted.
Author's Notes to Reviewer:
Thank for the review's great suggestion and help.
Reviewer 3 Report
Authors have made requisite changes and the quality is improved significantly.
Author Response
Reviewer:
Authors have made requisite changes and the quality is improved significantly.
Author's Notes to Reviewer:
Thank the reviewer's great help and suggestions.